# The Predictive Role of Quantity and Quality Language-Exposure Measures for L1 and L2 Vocabulary Production among Immigrant Preschoolers in Italy

**DOI:** 10.3390/ijerph20031966

**Published:** 2023-01-20

**Authors:** Arianna Bello, Paola Ferraresi, Maria Cristina Caselli, Paola Perucchini

**Affiliations:** 1Department of Education, Roma Tre University, Via Castro Pretorio 20, 00186 Roma, Italy; 2Institute of Cognitive Sciences and Technologies, National Research Council, Via Nomentana 56, 00161 Rome, Italy

**Keywords:** bilingual children, vocabulary production, MB-CDI, quantity and quality of language exposure, predictive role of language exposure

## Abstract

In this study, we investigated the lexical ability in L1 and L2 of 60 immigrant children who were 37 to 62 months old and exposed to minority languages (L1) and Italian (L2). Using the MacArthur–Bates Communicative Development Inventories, we measured children’s vocabulary production in L1 and L2. From interviews, we collected data on quantitative language exposure (parental input, child output, length of exposure to L2 at preschool, and parental oral fluency) and qualitative home-language exposure (HLE) practices (active, play, and passive) in L1 and L2. We conducted stepwise regression analyses to explore which factors predicted children’s vocabulary production in L1 and L2. The child’s chronological age and parental education were not predictors of vocabulary production. L2 parental input, L1 child output, and L1 HLE-active practices explained 42% of the variance in children’s L1 vocabulary production. L2 child output and L2 HLE-active practices explained 47% of the variance in children’s L2 vocabulary production, whereas length of L2 exposure in preschool was a predictor only when we included quantitative language-exposure factors in the model. The effects of the quantity and quality of language exposure on lexical ability among preschool immigrant children are discussed.

## 1. Introduction

The goal of the present study is to explore the relationship between language exposure and lexical ability in preschool children from immigrant families in Italy. Specifically, the study focuses on which language-environment factors explain the individual differences in vocabulary production between the children’s first language (L1, a minority language) and second language (L2, Italian).

The number of immigrant preschoolers is constantly increasing in Italy. In 2020/2021, immigrant preschoolers comprised 11.8% of the children attending kindergartens [1]. Immigrant children are exposed to L2 in preschool and in their social environments, whereas at home or in their ethnic communities they are exposed to L1, which is, in most cases, a minority language. The vast majority of these children grow up in socioeconomically disadvantaged circumstances [2].

An analysis of children’s expressive vocabulary in L1 and L2 at the ages of 3 to 4 years is crucial to understanding their word-combination acquisition [3] and morphosyntactic development [4,5]. However, children’s expressive vocabulary has been less investigated than their receptive vocabulary, although it is predictive of their literacy and academic success [6,7].

Several studies on linguistic development in immigrant children aged between 2 and 6 years of age, living in the United States and Europe, have shown that their expressive vocabulary lagged behind that of monolinguals in L1 [8,9] and L2 [3,8,9,10,11]. When comparing their expressive vocabulary in L1 and L2, studies reported mixed results: some authors documented lower competence in L1 than in L2 [3,9,12], and others found the opposite [11,13]. This incoherence of data likely occurred due to the language environments. For example, Kan and Kohnert studied adopted immigrant children and attributed the strong increase in L2 vocabulary to the L2-predominant context. Leseman and Ertanir studied Turkish immigrant children who lived in nearly monolingual L1 environments until the ages of 3 or 4.

Immigrant preschool children participating in studies of expressive vocabulary acquisition often come from diverse linguistic, familial, and environmental backgrounds. Therefore, consistent with the theoretical approach of the bioecological model of human development [14] and Hoff’s [15] claim that different environments produce individual differences in the rates of language acquisition, several studies explored expressive-vocabulary acquisition by considering a wide range of individual, linguistic, and environmental factors. For this reason, in the following paragraphs, we focus on the literature on language-exposure measures that influence children’s L1 and L2 preschool vocabulary production and discuss the still-debated question of which language-exposure variables are associated with children’s vocabulary production. A large number of studies have examined these associations using correlation analyses between measures of language exposure and vocabulary production (see Section 1.1). Only a limited number of studies, however, have used regression models to investigate the unique contributions of language-exposure measures, which we discuss in detail in Section 1.2.

### 1.1. Quantity and Quality of Language-Exposure Measures and Vocabulary Production

The term “language exposure” refers to the quantity and quality of language exposure [16]. Regarding the quantity of language exposure, researchers have distinguished between two measures of home exposure: “parental input”, which is the frequency with which children hear their parents using L1 and L2, and “child output”, which is the frequency with which children speak L1 and L2 with various family members (e.g., mothers, fathers, or siblings) or with all family members without distinction.

The parental input measure in L1 and in L2 was positively associated with children’s expressive vocabulary in L1 [9,17] and L2 [9,18,19]. Moreover, a negative correlation between L2 parental input and children’s L1 vocabulary production emerged, suggesting that the use of L2 by non-native speaking immigrant adults negatively affected their children’s L1 vocabulary acquisition [9,19].

The relationship between child output was consistent in children’s L1 but not L2 vocabulary production. Regarding L1, child output has been operationalized as the frequency with which a child uses L1 with their mother and siblings [19] and as the proportion of L1 spoken at home in a typical week [17]. Regarding L2, only the frequency with which the child uses L2 with mother, father and older siblings [19] and the speech the child uses to reply when parents speak in L2 and L1 [18] was positively associated but not the proportion of L2 the child uses in a typical week [17].

For L2, two quantitative measures of cumulative language exposure, which is defined as the amount of language exposure during the child’s life, were also investigated: “early day care entry” (the age of entry into day care) and “length of L2 exposure at preschool” (the lag between the age at which the child first enters day care/kindergarten and the age at which his/her language is measured). The quoted measures refer to the non-family context.

Positive strong correlations were found between children’s L2 vocabulary production and early day-care entry [13] and between children’s L2 vocabulary production and length of L2 exposure at preschool [3,20,21].

Although the quantity of language exposure was undoubtedly important for children’s language development during their preschool ages, research has suggested that the quality of home-language exposure (HLE) also played a crucial role. Specifically, researchers have measured the association between children’s vocabulary production and “combined” measures, considering an overall score of various types of practice, and “separate” measures, considering each type of practice separately. In the latter case, practices were classified as “high-level or active” (e.g., storytelling), “low-level or passive” (e.g., exposure to television), and specific linguistic interactions (e.g., types of child–caregiver conversations).

Pham and Tipton [17] documented, in each language, a strong positive association between combined measures in both L1 and L2, including the use of iPads, TV, movies, storytelling, and child’s vocabulary production in L1 but not in L2. Similarly, Paradis and Jia [20] found a positive association between L2 vocabulary and combined measures, including frequency of experience with media, books, and activities organized with peers in an average week. If separate measures are examined, the number of books at home was positively correlated with vocabulary in L2 [13,19] but not in L1 [13,19]. However, regarding L1, the number of books at home affected vocabulary in conjunction with individual and environmental variables [22]. The “reading input”— how frequently family members read to a child—was positively associated with vocabulary in L2 [11,19,23], but incoherent data were found in L1 (positive association, 11; lack of association, 24).

The effect of active and passive practices on vocabulary production has also been explored. Florit, Baracchetti, Majorano, and Lavelli [23] found that only active-level practices affected children’s expressive vocabulary in L2 at 2 years of age, even though L2 was less frequently used in immigrant families with low socioeconomic status (SES). Finally, Leseman [11] confirmed that specific activities, such as storytelling, talking about everyday facts, and reading books, enhanced L1 vocabulary and that home conversations, such as child–caregiver conversations during lunch/dinner, songs, and nursery rhymes, were crucial for promoting L2 vocabulary.

In summary, research has confirmed that for measures of home exposure, parental input is associated with children’s vocabulary production in L1 and L2, whereas children’s output plays a controversial role. The duration of preschool L2 exposure, as a cumulative measure of exposure, has been identified as a significant factor for vocabulary in L2. Thus, the quantitative exposure to language—at home and in an extrafamilial context—is a multifaceted and complex variable that still requires extensive study.

### 1.2. The Predictive Role of Language-Exposure Measures for Children’s Vocabulary Production

To the best of our knowledge, researchers have conducted only seven studies investigating the predictive effect of language-exposure measures on vocabulary production in L1 and L2 among preschool immigrant children. Regarding L1, two quantitative measures, parental input and child output, were identified as predictors of an expressive lexicon. Immigrant preschoolers with more L1 parental input [17] and less L2 parental input [21] demonstrated a larger L1 vocabulary. However, parental input was also a controversial factor, given its lack of a predictive role [13,20]. Other studies confirmed a positive effect of L1 child output [19,21] and a negative effect of L2 child output [21] on L1 vocabulary.

Considering the quality of language exposure at home, unlike combined measures [17], the number of questions mothers asked during shared book reading and maternal reading input were identified as predictors for children’s expressive L1 vocabulary [19]. Researchers have conducted few studies investigating the predictive role of individual and familial factors; therefore, their findings were partial or controversial. Children’s chronological ages and phonological memories were determinants [13], but the latter variable also seemed non-significant [17]. Familial factors, such as a mother’s years in the host country and the children’s school involvement, were found to be predictors for L1 vocabulary [19].

Regarding L2, the predictive role of both quantitative measures, parental input and child output, was debated. A positive influence of parental input [21,24] and child output [19,22] was found. A lack of their influence on child’s vocabulary also emerged [17,20,21]. Moreover, the predictor role of the length of L2 exposure in preschool [20,21] and early day-care entry [13] was strongly demonstrated.

Several qualitative language measures were predictors of L2 vocabulary: the combined measures of experience with media, books, and activities organized with friends in a typical week [20]; the number of questions asked during book reading interactions [19]; the number of books at home [13]; reading input, including the mother and father reading to the child and the availability of books [24]; the frequency of high-level practices promoting child–parent interaction (e.g., shared reading, storytelling, songs); and the duration of reading among immigrant toddlers of 30 months [23]. However, the combined qualitative measures of experience with an iPad, TV, movies, and storytelling [17] did not predict L2 vocabulary. Only one study demonstrated that maternal L2 oral fluency was not a predictor for L2 vocabulary [21]. Finally, individual factors, such as a child’s chronological age [17] and phonological memory [13,17], were determinants for L2 vocabulary, as were familiar factors related to integration (i.e., a mother’s years in the host country and school involvement) [19] Families’ SES was identified as a controversial predictor: SES was decisive in the study by Ertanir et al. [13] but not in the study by Rydland and Grøver [22]. Prevoo et al. [24] found that SES influenced maternal input and reading input directly and L2 vocabulary indirectly.

In sum, child output is a clear predictor of children’s L1 vocabulary production, whereas parental input is a controversial predictor of children’s L1 vocabulary production. The few explorations of qualitative language exposure and individual and familial factors do not provide an exhaustive picture of their influence on L1 vocabulary. The length of L2 exposure in preschool is a significant predictor of children’s L2 vocabulary production, unlike parental input and child output, which are more controversial predictors. The influence of qualitative language measures on L2 has been thoroughly investigated: the number of books, reading input, and child–caregiver interactions predicted children’s vocabulary in immigrant families. Phonological memory was also a predictor for L2 vocabulary, and SES indirectly predicted children’s L2 vocabulary.

### 1.3. Aims of the Study

The literature presented above shows a quite incomplete picture of the role of quantitative and qualitative language exposure in explaining individual differences in children’s vocabulary production. Researchers have conducted a limited set of regression studies to investigate the predictive effect of quantitative and qualitative language-exposure measures on L1 and L2, and separate qualitative measures have only recently been used to clarify their role fully. Therefore, the main aim of this study was to help verify the predictive effect of language exposure on L1 and L2 vocabulary by comparing quantitative and qualitative exposure measures, as well as individual and familial factors.

We analyzed current language exposure as parental input and child output in L1 and L2 and cumulative exposure as the duration of preschool L2 exposure. In addition, following the study of Florit et al. [23], we explored home language practices by distinguishing them according to the characteristics of communicative and linguistic interaction. Therefore, we distinguished HLE-active practices (e.g., listening to songs and nursery rhymes, telling stories and fairy tales, reading books with pictures) as the ideal context, in which high-quality input supports language acquisition, from HLE-passive practices (e.g., watching television), which lack high-quality communicative and linguistic interaction. We also included HLE-play practices (e.g., playing with dolls/puppets and playing with puzzles/constructions) as a context involving activities with objects that promote language acquisition. Finally, we explored demographic variables as individual (e.g., chronological age) and familial (e.g., parental education) factors.

Specifically, the aims of this study were the following:

1. A preliminary aim was to investigate the association of quantitative and qualitative measures of language exposure with children’s vocabulary production in both languages. Regarding L1, we expected correlations with parental input [9,17] and child output [17,19], but we did not make predictions for length of preschool L2 exposure and parental L2 oral fluency (i.e., oral language competence), due to the lack of previous studies. We also expected correlations with HLE-active practices [11,17], but we did not make predictions for HLE-play and HLE-passive practices, due to the lack of previous studies. In line with other studies [25,26], we expected an association with maternal and paternal educational levels.

Regarding L2, we assumed correlations with parental input [9,18,19] and length of preschool L2 exposure [3,20,21], based on previous studies, but we did not make predictions for child output because the available data were not consistent [17,19]. We expected associations with HLE-active practices [20] but not with HLE-passive practices, as documented in immigrant toddlers [23], and made no prediction for HLE-play due to the lack of previous studies. We also assumed an association with familial educational levels for L2 [25,26].

2. The main aim of the present study was to investigate which factors explained the individual differences in children’s vocabulary production in L1 and L2, by comparing the predictive role of quantitative and qualitative exposure language factors, as well as individual and familial factors. To pursue that aim, we conducted regression analyses, including consideration of the several factors separately. Regarding L1, we assumed a predictive role of child output [19,21,22], but we did not make a prediction for parental output, due to a lack of univocal data [13,20], or for the length of preschool L2 exposure or L2 parental oral fluency, due to the lack of data. We did not make a prediction for qualitative measures, due to inconsistent [19] or controversial data [17]. To our knowledge, no data were available on the predictive effect of the maternal educational level on L1 vocabulary.

Regarding L2, we assumed a predictive role of the length of preschool L2 exposure [18,19], but we did not make a prediction for parental input or child output, due to incoherent data [17,19,20,21,22,24]. We assumed a predictive role of HLE-active practices, due to strong data [19,23]. To our knowledge, no data were available on the predictive role of HLE-passive practices or HLE-play practices for preschool immigrant children. Therefore, we aimed to fill the gap in the literature.

Because there was no evaluation of the Italian tool for vocabulary production for 3-to-5-year-old immigrant children exposed to several L1s, we adopted the MacArthur–Bates Communicative Development Inventories (MB-CDI) questionnaire for bilinguals. This questionnaire is deemed suitable for children who are 2 to 3 years old, and it includes a consistent list of 100 words to use for L1 and L2. Furthermore, the use of this tool is justified, because among these immigrant preschoolers, the ages of systematic exposure to L2 was lower than their chronological ages and fell within the age range within which the questionnaire is administered to monolingual children. As Bello et al. demonstrated [3], at preschool age, reliable measures of expressive vocabulary among immigrant children were separate L1 and L2 vocabularies.

## 2. Materials and Methods

### 2.1. Participants

The participants were 60 children (34 females) with a mean chronological age of 49.92 months (SD, 7.10 months) born of immigrant parents in northern Italy (see Table 1). They used their L1 at home and were learning Italian as L2, primarily in kindergartens.

The exclusion criteria for the selection of children were the following: <6 months of exposure to Italian at school; having one native-speaking L2 parent; premature birth; being twins; and having suspected or overt neurodevelopmental disorders.

The immigrant children were exposed to 15 heritage languages at home: Moldovan, Albanian, Arabic (used in Algeria, Morocco, and Tunisia), Spanish (used in Argentina, Colombia, Santo Domingo, Paraguay, and Peru), English (used in Nigeria), French (used in Cameroon and The Ivory Coast), Bangla, Chinese, Amharic, Bambara, Fante, Romanian, Tagalog, Twi, and Wolof. Immigrant families came from 21 countries. Most had lived in Italy for >5 years, as had 57 of the mothers (95.0%) and 55 of the fathers (98.2%). The mean proportion of the current amount of exposure for L1 was 33.5% (SD, 11.4%; range, 8–57%); for L2, it was 66.5% (SD, 10.8%; range, 43–92%) in a typical child’s week (for more detail, see Linguistic Biography, Section 2).

Regarding families’ education levels, the mothers had attended school for a mean of 12.27 years (SD, 3.63 years) and the fathers had attended school for a mean of 11.82 years (SD, 3.45 years). All of the families signed written informed consent forms, in line with the procedures approved by the Ethics Committee of Roma Tre University.

### 2.2. Procedures

We collected data in a single 1 h meeting with the immigrant parents. Only mothers (78%), only fathers (10%), or both parents (12%) participated in these meetings. All parents participated without the need for any linguistic mediation. During the meetings, parents responded to the semi-structured interview (the “linguistic biography”) and then completed the MB-CDI questionnaire.

### 2.3. Tools

#### 2.3.1. Linguistic Biography

The semi-structured interviews were termed the “linguistic biography” and were a modified version of the original interview by Onofrio et al. [27]. A description of the questions and measures obtained during this study is set out below.

Section 1: Child and parent information. This section included questions relating to the child’s health status (e.g., term birth, health problems), the parents’ educations, and their time living in Italy. We determined the parents’ education levels based on their years of school attendance.

Section 2: Language environment. This section included several questions regarding the exposure to and use of L1 and L2 by parents and children, considering the quantity and quality of exposure to languages.

First, we used a child’s attendance at an Italian infant day care and/or kindergarten to calculate the length of exposure to L2, expressed in months, as the current age minus the age of first exposure to Italian in day care/kindergarten.

Second, we asked about the frequency of L1 and L2 use at home, when parents spoke with their child (parental input), and when the child answered them (child output). We measured parental input and child output in L1 and L2 on a 3-point scale (1 = never; 2 = rarely or sometimes; 3 = often or always). For the analysis of parental input and child output, we calculated the mean of the scores for mothers and fathers in L1 and L2.

Then, we estimated the current amount of exposure to L1 and L2 using the following question: “Since the child has been going to school, how has the child’s week been organized, considering 12 h of wakefulness?” Together with the parents, we reconstructed a typical child’s week to determine the length of time and the languages used at various times of the day during the activities carried out with the people with whom the child had regular contact. We computed a rough estimate of the amount of exposure to each language as a percentage (%) of the exposure to L1 and L2 in a typical week to describe the characteristics of the sample.

Finally, we analyzed the use of L1 and L2 in typical routines in the previous month of the child’s life, including child–parent interaction. Specifically, the routines were (1) watching television, (2) listening to songs and nursery rhymes, (3) telling stories and fairy tales, (4) reading books with pictures, (5) playing with dolls/puppets, and (6) playing with puzzles/constructions. We asked parents to indicate the frequency of use of L1 and L2 in each routine. We recorded their answers on a 4-point scale for each language (1 = never, 2 = twice a month, 3 = once or twice a week, 4 = every day). In the analyses, we used as measures the score for routine 1; the average score for routines 2, 3, and 4; and the average score for routines 5 and 6. We calculated these scores for each of the two languages (L1 and L2). We grouped the routines based on the characteristics of communicative and linguistic interaction, following Florit et al. [23]. Therefore, we considered watching TV an activity with passive exposure to language. We considered activities 2, 3, and 4 active communicative and linguistic interactions with caregivers, and we considered play (routines 5 and 6) as more focused on the action and the object than the routines in 2, 3, and 4.

At the end of the interviews, a native Italian-speaking interviewer estimated the parents’ oral fluency in Italian on a 4-point scale (1 = a low degree, 2 = partial, 3 = a high degree, 4 = a very high degree). For the analysis, we combined the first two responses as “low proficiency level” and the third and last responses as “high proficiency level”.

#### 2.3.2. The MB-CDI Short-Form Questionnaire for Bilinguals

As the children in the sample used 15 different L1s and no MB-CDI questionnaires were available for these languages, we used a new version of the MB-CDI that is usable with any L1. It was specifically designed for immigrant parents of preschool children [28]. This version is reliable because the obtained measures in L1 correlated with the corresponding measures estimated by the short form of the MB-CDI questionnaires that are adapted for different L1s [28].

In this study, we analyzed only the section “vocabulary list”. That section presents the original list of 100 Italian words to which a column referring to L1 words was added. We asked the parents whether their child produced the Italian words and whether they produced the L1 equivalents to the Italian words. We obtained two vocabulary measures (giving each word a score of 1): vocabulary raw score in L2 (the sum of the Italian words produced) and vocabulary raw score in L1 (the sum of the words produced in L1).

### 2.4. Plan of the Analyses

Regarding the main aim, we analyzed the data using stepwise regressions in order to explore which predictors explained individual differences in the lexical abilities of immigrant preschool children. The predictors considered were both individual (the child’s chronological age) and familiar (the mother’s and father’s educations), and related to quantitative (parental input and child output) and qualitative measures of language exposure (HLE practice), while the outcome variables were the vocabulary production in L1 and L2 detected through the MB-CDI questionnaire. Before running the regression models, we conducted correlations between predictor variables in order to see if they were correlated (0.5–1, moderate to strong) by Spearman’s Rho test (preliminary aim).

In particular, we performed two stepwise regressions, one for vocabulary production in L1 and one for vocabulary production in L2. In the two regression models, we first entered demographic measures, such as the child’s chronological age and the educational levels of the father and mother (step 1). In step 2, we subsequently added quantitative measures of language exposure, such as the length of preschool L2 exposure, L1 and L2 parental input, and the parents’ L2 oral fluency. Finally, in step 3, we added L1 and L2 child output as predictors of children’s vocabulary. Although the parental input and child output variables in L1 and in L2 correlated with each other, we decided to include them in steps 1 and 2 for each regression model in order to compare the facilitation and inhibition roles that they may play on children’s expressive vocabulary for each language. Then, we ran six separate stepwise regressions, one for each HLE practice, as active, play, and passive, on the children’s vocabulary production in L1 as well as in L2. In these models, we inserted the demographic characteristics at the first step; in the second step, we inserted the measures that were significant in the two previous regressions on the quantitative factors specific to each language. In the model of the children’s L1 vocabulary production, we included the L2 parental input and the L1 child output. In the model of the children’s L2 vocabulary production, we included only the L2 child output. In the third step, we inserted, for each model separately, the three HLE practices (active, play, and passive) in L1 and in L2.

## 3. Results

### 3.1. Descriptive Analysis

Table 1 displays the descriptive data of linguistic outcome variables (children’s vocabulary production in L1 and L2) and predictors as demographic measures and measures of quantity and quality of language exposure.

As shown in Table 1, on average, the children had a more advanced vocabulary in L2 than in L1: they produced more than twice the words in Italian in L1. Six of the children did not produce words in their native language, whereas only one child did not produce any words in L2. We observed high inter-individual variability for both L1 and L2.

The parental input score was very similar for L1 and L2, while the mean child output score was higher in L2 than in L1. The length of L2 exposure at preschool averaged 28.20 months, with consistent inter-individual variability. In addition, 73.30% of the parents interviewed had good L2 (Italian) oral fluency, producing few grammatical errors and using an appropriate lexicon, and they understood most of what was said; 26.70% of the parents had poor oral Italian proficiency, producing grammatical errors with a limited Italian vocabulary.

All HLE practices were more frequently used in L2 than in L1. In L2, the passive practices (e.g., watching TV) and play were used more with respect to the active practices (e.g., singing and nursery rhymes, storytelling, and book reading). In L1, TV viewing was frequently adopted, and active practice involving songs, nursery rhymes, storytelling, and book reading were used.

### 3.2. Correlations

Table 2 displays the correlations between children’s vocabulary production in L1 and L2 and demographic variables, quantitative measures, and qualitative measures of linguistic exposure.

No correlations emerged with demographic variables (child’s chronological ages and mother’s and father’s educations), either for L1 or for L2 (see Table 2).

The child’s L1 vocabulary production correlated positively and strongly with the L1 child’s output and moderately with L1 parental input; in contrast, it correlated negatively and moderately with L2 parental input and with the L2 child’s output. No correlations emerged with the length of preschool L2 exposure or parental L2 oral fluency.

The child’s L2 vocabulary production correlated positively and strongly with the length of preschool L2 exposure and moderately with L2 child output. No correlation emerged with L1 child output or L1 parental input, or with parental L2 oral fluency.

The child’s L1 vocabulary production correlated positively and moderately with qualitative measures of language exposure, as L1 HLE-active practices and L1 HLE-play practices. We found no correlation with L1 HLE-passive practices or with HLE practices in L2. The child’s L2 vocabulary production also correlated positively and moderately with HLE-active practices and HLE-active practices in L2. No correlation emerged with L2 HLE-passive or with HLE practices in L1.

### 3.3. Regressions

Table 3 displays a summary of the stepwise regression analyses with steps 1, 2, and 3 for children’s vocabulary production in L1 and L2.

The results of the first regression analysis showed that no demographic factors contributed to children’s vocabulary production in L1 (step 1); on the other hand, the L2 parental input negatively contributed in step 2, with an R2 of 0.299. In step 3, the last factor still negatively contributed, together with L1 child output, with an increase of R2 to 0.400 (see Table 3).

The results of the second regression analysis showed that no demographic factor contributed to children’s vocabulary production in L2 (step 1); on the other hand, the length of L2 exposure at preschool contributed in the second step with an R2 of 0.259, but not in step 3. When child output in L1 and L2 were added (step 3), only the L2 child output was significant, with an R2 of 0.371 (see Table 3).

Table 4 displays a summary of the three stepwise regression analyses with step 3, including separate practices (active, play, and passive) for children’s vocabulary production in L1. Looking at children’s L1 vocabulary, the results of the regressions showed that no demographic factor contributed (step 1). However, in step 2, L2 parental input negatively contributed, while L1 child output positively contributed, with an explained variance of 35%. When active practices in L1 and L2 were added (step 3 HLE-active), R2 increased to 0.422 and was determined by child output and HLE-active practices in L1. In the second model (step 3 HLE-play) and in the third model (see step 3 HLE-passive), R2 was not significant. Thus, in both L1 and L2, play and passive practices did not have predictive roles for children’s vocabulary production in L1.

Table 5 displays a summary of the three stepwise regression analyses with step 3, including separate practices (active, play, and passive) for children’s vocabulary production in L2. Regarding children’s L2 vocabulary, the results of the regressions showed that no demographic factors contributed (step 1). However, L2 child output contributed in the second step, with an R2 of 0.331. When we added HLE-active practices in L1 and L2 (step 3 HLE-active), L2 child output was still significant, together with HLE-active practices in L2, with an R2 of 0.470.

In the second model (step 3 HLE-play; Table 5) and in the third model (step 3 HLE-passive; Table 5), R2 was not significant. Thus, play and passive practices in L1 and L2 did not have predictive roles in children’s vocabulary production in L2.

## 4. Discussion

Our main aim in this study was to investigate in 4-year-old immigrant children—all of them attending kindergarten—the predictive role of language exposure on vocabulary skills in both L1 and L2, considering quantitative and qualitative measures. In particular, we analyzed the frequency with which the two languages are used at home by parents (parental input) and children (child output). We also considered cumulative exposure as the length of L2 exposure (i.e., the child’s current age minus the child’s age at first exposure to Italian at nursery/kindergarten). In addition, we examined the predictive role of individual factors (e.g., the child’s chronological age) and familial factors (e.g., parental education). Finally, we explored the predictive role on child’s lexical abilities of home-language practice (HLE), both active (e.g., listening to songs and nursery rhymes, telling stories, telling fairy tales, and reading books with pictures) and passive (e.g., watching television), as well as play activities (e.g., playing with dolls or puppets and playing with puzzles or construction sets).

Our results found no association between children’s lexical abilities and the educational level of the mothers and fathers. These results are not in line with those reported in previous studies, where parental educational level [26] or maternal educational level [29] was associated with L2 vocabulary. We may attribute this discordance to the characteristics of our samples: in the previous studies, immigrant preschoolers belonged to differentiated groups of families with low or high education levels. On the contrary, the children in our sample belonged to families with heterogeneous educational levels. This higher variability may have masked the effects reported in previous studies of these variables on children’s lexical skills in the two languages. In accordance with the results reported by Sorenson, Duncan, and Paradis [21], our data indicated no association between children’s vocabulary in L1 and L2 and parental L2 oral proficiency. Our results also confirmed that vocabulary skills in both languages were not related to children’s chronological ages, but above all were related to language-exposure variables [30]. In the following paragraphs, we discuss in more detail the relationship between the variables and children’s expressive vocabulary in L1 and L2.

### 4.1. Children’s Vocabulary Production in L1 and Language-Exposure Measures

The preliminary aim of this study concerned the association between quantitative and qualitative measures of exposure to language and children’s vocabulary production. In quantitative measures, we found a strong positive correlation with the L1 child output and a moderate positive correlation with the L1 parental input. We also found a moderate negative correlation of the L1 child vocabulary production with both the L2 parental input and the L2 child output. As for qualitative measures, we found a positive moderate correlation with both the L1 HLE-active practices and the L1 HLE-play practices, and a lack of correlation with the L1 HLE-passive practices (e.g., watching TV).

These correlational results were specified in regression analysis, through which we investigated the predictive role of exposure factors (our main aim). When in the regression analysis we entered the quantitative measures exclusively, only two factors emerged as predictors in the final model, explaining 40% of children’s L1 vocabulary variance: the child output of L1, which positively predicted the L1 vocabulary, and the parental input of L2, which negatively predicted the L1 vocabulary. If, in the regression analysis, we separately added the three HLE-practices (i.e., active, passive, and play; see Table 4), then the parental input of L2 did not obtain the predicted result. The final model explained 42% of the variance in children’s L1 vocabulary through the contribution of two factors as L1 child output and L1 HLE-active practices. L1 HLE-play and L1 HLE-passive practices were not sufficiently significant to explain the inter-individual variability. Our data concerning the predictive role of the L1 child output in L1 children’s vocabulary production confirmed previous data [19,22], and our data regarding the predictive role of L1 HLE-active practices extended previous studies [11,19] that considered different measures of exposure quality. Active practices offer a richness of language input from parents, in terms of lexical diversity and grammatically rich constructions, among others [21], that promote vocabulary acquisition.

In sum, the development of children’s vocabulary production in L1-minority contexts is determined by the children’s use of L1 at home and by the parental use of interactive practices, such as listening to songs and nursery rhymes, telling stories and fairy tales, and reading books with pictures, which are all suitable for stimulating the vocabulary. Parental input of L2 at home negatively interferes with the child’s L1 vocabulary development if only quantitative measures are considered; otherwise, if both quantitative and qualitative measures are considered, parental L2 input at home is not found to be associated with children’s L1 vocabulary. Early systematic exposure to L2 in the Italian educational context does not appear to interfere with children’s L1 vocabulary production.

### 4.2. Children’s Vocabulary Production in L2 and Language-Exposure Measures

Regarding the association and predictive role of language exposure on children’s L2 vocabulary, our results confirmed and extended the previous data. We found a strong positive correlation with the length of preschool L2 exposure and a moderate positive correlation with the L2 child output, L2 HLE-active practices, and L2 HLE-play.

These correlational results were specified by regression analysis, through which we investigated the predictive role of exposure factors. As for the predictive role of the exposure factors, if in the regression analysis (see Table 3), we entered the quantitative measures, the length of L2 exposure at preschool was a predictor when the L2 child output was not included. The predictive role of early systematic exposure to L2 by native speakers (i.e., teachers and pupils) in the educational setting was in line with previous data [20,21]. However, when the use of L1 and L2 by the child with family members were included in the regression, only one factor appeared to be a predictor in the final model—the L2 child output—explaining 37% of the variance. These last data were consistent with data of the previous study [19,22].

If, in the regression analysis, we separately added the three practices (active, passive, and play; see Table 5), only the model considering the L2 HLE-active practices had significant results. This final model explained 47% of the variance in children’s vocabulary production with two factors: L2 child output and the use of HLE-active practices in L2.

Our data on the predictive role of L2 active practices extend the data available in the literature, where several separate measurements were found to be predictors: maternal practices [19], number of books [13], and reading input [24]. Our data confirmed the results of Florit et al. (2021), who found that the frequency and the duration of shared reading, storytelling, and songs, but not watching TV, were predictors of L2 vocabulary in 30-month-old immigrant children. In addition, for L2, active practices at home provided the ideal context for vocabulary learning, demonstrating that the richness of child–caregiver language interaction induced by the use of books, stories, and songs promotes lexical development even in bilingual contexts [23].

In our sample of children aged from 3 to 5 years, watching TV occurred at home with high frequency, but this practice did not affect children’s L2 vocabulary. We suggest that, given the age range and the L2 vocabulary levels of our sample, the exposure to L2 outside the communicative–linguistic context was not adequate to promote language development.

The lack of a relationship between parental L2 oral fluency and L2 vocabulary is in line with the literature that documented the influence of this factor only on more complex linguist abilities, such as morpho-syntactic skills [30,31].

In sum, the ability of children to use L2, rather than being systematically exposed to L2 by parents at home, is a determinant predictor of L2 vocabulary production in immigrant preschoolers. The development of an L2 (Italian) vocabulary is determined by both the child output and the use of active practice in L2. Parental oral fluency in L2 is not determined to predict L2 vocabulary, while early systematic exposure to L2 by native speakers (i.e., teachers and pupils) influences children’s L2 vocabulary acquisition only when the parental use of active practices is minimally adopted.

### 4.3. Strengths and Limitations of the Study

The present study on immigrant preschoolers contributes to the literature on initial linguistic development in three ways. First, immigrant children are a very heterogeneous population, and there are no exhaustive data on which factors affect the development of their vocabulary production. This population requires deep analysis, as it is growing in the context of school participation and might cause difficulties in children’s schooling. This study recruited children at schools to provide a picture of the real school population, at least with reference to the geographical and cultural contexts in which the survey took place.

Second, we examined the vocabulary skills of immigrant preschool children attending kindergarten. It is crucial to understand the quantitative and qualitative factors of linguistic exposure that affect the development of L2 (Italian) so that teachers can promote L2 learning through targeted educational activities that promote language interaction.

Third, immigrant children aged 3–5 years have achieved levels of productive vocabulary measured by the MB-CDI that are more than twice as high in L2 as in L1. It is, therefore, important to know what factors of linguistic exposure predict the development of both languages to provide correct information to immigrant families so that they can promote the learning of both L1 and L2.

Some limitations of this study need to be highlighted. Although the sample of children is representative of the variety of immigrant children present in Italian schools, that sample was very heterogeneous for native languages, for systematic exposure at school to L2, and for the current amount of exposure to L1 and L2. Therefore, because the phenomenon of bilingualism is multifaceted, the results obtained in this study are not easily comparable with those of other studies.

Another limitation of this study is that we detected vocabulary skills through a list of 100 words in L1 and L2 (i.e., the MB-CDI short form). On one hand, this selected list of words may underestimate the abilities of children in terms of a ceiling effect, although given the wide variability, we were able to identify individual differences. We had planned to use both the indirect measure of vocabulary assessment and the direct measure of vocabulary skills through lexical naming. However, due to the coronavirus pandemic and the closure of kindergartens, this was impossible.

The last limitation of the study is that, we mainly assessed, as factors of linguistic exposure, current quantitative measures of exposure and only a cumulative measure for the time of systematic exposure to L2 at school. At present, there is no valid instrument to measure cumulative language exposure and it is very complex to operationalize cumulative language exposure, even if it represents the most appropriate measure in conditions of bilingualism [32]. With regard to the qualitative factors of linguistic exposure, we examined active practices involving the use of narrative, reading books, and playing with puppets and construction sets, even though these practices are more customary in Western cultural contexts and are subject to specific cultural influences.

## 5. Conclusions

On the application level, the findings of this study provided evidence about which factors, such as child output and HLE-active practices, predict the development of a child’s vocabulary production for both L1 and L2, and which factors should be promoted in educational and similar contexts. The attrition effect of L1-minority and the dominance of L2-Italian in preschool immigrant children emphasizes the need to encourage educational practices in families that promote L1 and L2. Immigrant parents do not seem to be aware of the importance of maintaining L1 and, therefore, they should be encouraged to reflect on educational practices that most stimulate language interactions with cascading effects on children’s language acquisition. The privileged role of teachers could include accompanying parents in bilingual education. Teachers could promote the use of an easy-to-apply tool, such as the MB-CDI questionnaire, to initially test language acquisition in children exposed to two languages as immigrant preschoolers.

## Figures and Tables

**Table 1 ijerph-20-01966-t001:** Descriptive data (mean; percentage; standard deviation and range) of linguistic outcome variables (children’s vocabulary production in L1 and L2) and predictor variables (demographic, quantity of linguistic exposure, and quality of linguistic exposure).

Linguistic Outcome Variables	M (%)	SD	Range
Children’s vocabulary production in L1	37.9	28.3	0–100
Children’s vocabulary production in L2	85.6	25.7	0–100
Predictor variables			
Demographic			
Child’s chronological age (months)	49.92	7.10	37–67
Mother’s education (years)	12.27	3.63	4–18
Father’s education (years)	11.82	3.45	5–18
Quantity of language exposure			
L1 parental input	2.50	0.54	1–3
L1 child output	2.12	0.67	1–3
L2 parental input	2.47	0.52	1–3
L2 child output	2.83	0.42	1–3
Length of L2 exposure at preschool (months)	28.20	13.43	6–53
Quality of language exposure			
L1 HLE-active	1.86	0.84	1–4
L1 HLE-play	1.60	1.08	1–4
L1 HLE-passive	1.95	1.25	1–4
L2 HLE-active	2.10	0.91	1–4
L2 HLE-play	3.08	1.03	1–4
L2 HLE-passive	3.83	0.53	1–4

Note: We measured children’s vocabulary production on a 100-point scale, chronological age in months, parents’ educations in years, parental input and child output on 3-point scale, length of L2 exposure at preschool in months, and home language routines (HLE) on a scale 1–4.

**Table 2 ijerph-20-01966-t002:** Correlations between children’s vocabulary production in L1 and L2 demographics, quantity of language exposure measures, and quality of language exposure measures.

Correlations	Children’s Vocabulary Production in L1		Children’s Vocabulary Production in L2	
	rs	*p*	rs	*p*
Child’s chronological age	0.159	0.226	0.221	0.098
Mother’s education	0.172	0.189	0.168	0.212
Father’s education	0.000	0.999	0.172	0.189
L1 parental input	0.329	**0.010**	−0.174	0.195
L1 child output	0.513	**0.001**	−0.119	0.379
L2 parental input	−0.438	**0.001**	0.245	0.066
L2 child output	−0.388	**0.002**	0.357	**0.006**
Length of L2 exposure at preschool	0.142	0.277	0.512	**<0.001**
Parental L2oral fluency	0.162	0.216	0.155	0.250
L1 HLE-active	0.490	**<0.001**	−0.033	0.808
L1 HLE-play	0.394	**<0.002**	−0.142	0.294
L1 HLE-passive	0.254	0.050	−0.120	0.376
L2 HLE-active	−0.026	0.843	0.357	**0.006**
L2 HLE-play	0.157	0.230	0.374	**0.004**
L2 HLE-passive	−0.191	0.145	0.041	0.763

Note: Spearman’s correlations (rs). Significant results are in bold.

**Table 3 ijerph-20-01966-t003:** Summary of the stepwise regression analyses with steps 1, 2 and 3 for children’s vocabulary production in L1 and L2.

**Children’s Vocabulary Production in L1**									
	Step 1			Step 2			Step 3		
	β	t	*p*	β	t	*p*	β	t	*p*
Child’s chronological age	0.095	0.684	0.497	−0.047	−0.320	0.752	−0.065	−0.451	0.654
Mother’s education	0.185	1.014	0.315	0.282	1.606	0.102	0.197	1.198	0.237
Father’s education	−0.049	−0.300	0.765	−0.049	−0.300	0.765	−0.091	−0.587	0.560
	F(3, 52) = 6.15, *p* = 0.608; R^2^ = 0.034								
Length of L2 exposure at preschool				0.120	0.818	0.417	0.222	1.497	0.141
L1 parental input				0.033	0.212	0.833	−0.077	−0.488	0.628
L2 parental input				−0.551	−0.3.291	**0.002**	−0.332	1.855	**0.070**
Parental L2 oral fluency				0.045	0.324	0.748	0.059	0.446	0.658
				F(4, 48) = 4.530, *p* = 0.003, R^2^ = 0.299					
L1 child output							0.308	1.855	**0.070**
L2 child output							−0.205	−0.1286	0.205
							F(2, 46) = 3.878, *p* = 0.028, R^2^ = 0.400		
**Children’s Vocabulary Production in L2**									
	Step 1			Step 2			Step 3		
	β	t	*p*	β	t	*p*	β	t	*p*
Child’s chronological age	0.139	0.982	0.331	−0.023	−0.151	0.881	0.092	0.608	0.546
Mother’s education	0.176	0.978	0.333	0.024	0.134	0.894	0.061	0.351	0.727
Father’s education	0.044	0.244	0.808	−0.015	−0.091	0.928	−0.009	−0.057	0.955
	F(3, 49) = 1.175, *p* = 0.329; R^2^ = 0.067								
Length of L2 exposure at preschool				0.390	2.475	**0.017**	0.206	1.260	0.214
Parental input in L1				0.104	0.640	0.525	0.072	0.435	0.666
Parental input in L2				0.227	1.313	0.196	0.027	0.150	0.882
Parental L2 oral fluency				0.130	0.846	0.402	0.206	1.260	0.214
				F(4, 45) = 2.917, *p* = 0.031; R^2^ = 0.259					
Child output in L1							0.067	0.384	0.703
Child output in L2							0.454	2.724	**0.009**
							F(2, 43) = 3.822, *p* = 0.030; R^2^ = 0.371		

Note: *p* value for inclusion in the model was set at 0.1. Significant results are in bold.

**Table 4 ijerph-20-01966-t004:** Summary of the three stepwise regression analyses with step 3, including separate practices (active, play, and passive) for children’s vocabulary production in L1.

Children’s Vocabulary Production in L1															
	Step 1			Step 2			Step 3 HLE-active			Step 3 HLE-play			Step 3 HLE-passive		
	β	t	*p*	β	t	*p*	β	t	*p*	β	t	*p*	β	t	*p*
Child’s chronological age	0.095	0.684	0.457	0.057	0.476	0.636	0.073	0.625	0.535	0.055	0.459	0.648	0.070	0.567	0.573
Mother’s education	0.185	1.014	0.315	0.227	1.421	0.161	0.084	0.501	0.619	0.178	1.102	0.276	0.235	1.458	0.151
Father’s education	−0.096	−0.529	0.599	−0.070	−0.455	0.651	−0.123	−0.822	0.415	−0.086	−0.571	0.570	−0.089	−0.550	0.585
	F(3, 52) = 6.15, *p* = 0.608, R^2^ = 0.034														
L2 parental input				−0.343	−2.292	**0.026**	−0.202	−1.297	0.201	−0.379	−0.2377	0.021	−0.304	−1.920	0.061
L1 child output				0.311	2.181	**0.034**	0.272	1.156	**0.056**	0.253	1.715	0.093	0.299	2.073	0.044
				F(2, 50) = 12.129, *p* < 0.001, R^2^ = 0.350											
L1 HLE-active							0.361	2.437	**0.019**						
L2 HLE-active							−0.074	−0.604	0.548						
							F(2, 48) = 2.983, *p* = 0.060, R^2^ = 0.422								
L1 HLE-play										0.136	0.958	0.343			
L2 HLE-play										0.225	1.827	0.074			
										F(2, 48) = 1.989, *p* = 0.148, R^2^ = 0.399					
L1 HLE-passive													0.136	1.079	0.286
L2 HLE-passive													0.024	0.189	0.851
													F(4, 48) = 3.945, *p* = 0.586, R^2^ = 0.365		

Note: Significant *p*-values are in bold.

**Table 5 ijerph-20-01966-t005:** Summary of the three stepwise regression analyses with step 3, including separate practices (active, play, and passive) for children’s vocabulary production in L2.

Children’s Vocabulary Production in L2															
	Step 1			Step 2			Step 3 HLE-active			Step 3 HLE-play			Step 3 HLE-passive		
	β	t	*p*	β	t	*p*	β	t	*p*	β	t	*p*	β	t	*p*
Child’s chronological age	0.139	0.982	0.331	0.192	1.576	0.122	0.238	2.134	0.038						
Mother’s education	0.176	0.978	0.333	0.159	1.036	0.306	0.004	0.024	0.981						
Father’s education	0.044	0.244	0.808	0.016	0.103	0.919	0.019	0.134	0.894						
	F(3, 49) = 1.175 *p* = 0.329; R^2^ = 0.067														
L2 child output				0.518	4.349	**<0.001**	0.555	4.609	**<0.001**	0.479	3.343	0.002	0.533	4.348	<0.001
				F(1, 48) = 18.912, *p* < 0.001 R^2^ = 0.331											
L1 HLE-active							0.128	0.922	0.361						
L2 HLE-active							0.341	2.886	**0.006**						
							F(2, 46) = 6.039, *p* = 0.005,R^2^ = 0.470								
L1 HLE-play										0.060	0.420	0.677			
L2 HLE-play										0.189	1.444	0.155			
										F(2, 46) =1.086, *p* = 0.346 R^2^ = 0.361					
L1 HLE-passive													0.036	0.291	0.773
L2 HLE-passive													0.141	1.087	0.282
													F(2, 46) = 0.597, *p* = 0.555, R^2^ = 0.348		

Note: *p* value for inclusion in the model was set at 0.1. Significant results are in bold.

## Data Availability

The datasets used and analyzed in the current study are available from the corresponding author upon reasonable request.

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
