# Peer review of "The Predictive Role of Quantity and Quality Language-Exposure Measures for L1 and L2 Vocabulary Production among Immigrant Preschoolers in Italy"

_ijerph, 2023, doi:10.3390/ijerph20031966_

Round 1

Reviewer 1 Report

Methodology

l. 175-177: Why distinguish input and output in L1, but not in L2 (“cumulative exposure”)? Isn’t it likely that the proportion of input and output of L2 are just as relevant as in L1?

l.204: Why is there a lack of data regarding parental oral fluency? Could this not have been determined for each parent?

l. 206-207: “... no data was available on the predictive effect of maternal educational level on L1 vocabulary.” I’m not sure I understand this sentence. Do you mean there are no previous studies identifying such an effect? The data collected for this study should allow you to identify such an effect, shouldn’t it?

l. 272-273 “12 hours of wakefulness” I doubt 12 hours of wakefulness is a realistic estimate in all cases, and in terms of methodology it would have been better to ask how many hours the child is awake and then determine the proportion of L1 and L2 exposure. If day care takes up 8 hours per weekday, there is a considerable difference between a child that spends 4 hours awake and interacting with its parents and a child that spends 6 hours doing the same. In other words, this is so rough an estimate that it is likely to skew the results.

l. 306-307: “We asked the parents whether their children produced [the words in L1 and L2]” This is an extremely unreliable methodology, bordering on hearsay. Rather than asking parents to guess which words their children know in which of the two languages, why not test the children? Parents cannot are unlikely to know exactly which words their children do or don’t know, especially in L2. (In lines 587-590 it is acknowledged that a direct assessment, lexical naming, would have been better. Whilst the covid pandemic certainly made this kind of data collection more difficult, it would be important for such direct assessment of children’s vocabulary to be carried out and the results to be included in this study before publication.

l. 579-580: “The sample of children was ... very heterogeneous for native language”: The article acknowledges this as a weakness. It is indeed highly problematic to make no difference between L1 languages that are typologically extremely different from L2 on the one hand (e.g. Arabic, Chinese), and other L1 languages that are lexically and structurally quite similar to Italian (e.g. Spanish, Romanian). Children with the latter as heritage language are likely to increase their L1 and L2 vocabulary simultaneously during both L1 and L2 activities, whereas this will not be the case for languages such as Chinese and Arabic. This is a factor that should have been taken into account.

Discussion

l. 498-499: “Finally, our data on the lack of a predictive role for early and systematic exposure to L2 in the preschool is also original.” This sentence needs to be more precise. As I understand it, early and systematic exposure to L2 in the preschool does predict better L2 vocabulary. What it doesn’t influence is the child’s range of expressive vocabulary in L1. So you should probably write something like “Finally, the fact that early and systematic exposure to L2 does not have a significant influence on L1 vocabulary is also an original finding of this study.”

l.546-549: “Since play with puppets or construction sets is not properly representative of the immigrant parents’ cultural background, we hypothesize that it may induce the enactment of communicative-linguistic strategies not suitable for promoting L2 development.” This is a very strong generalization, especially as we are dealing with at least 15 different cultural backgrounds. In any case, play activities DO contribute to a larger vocabulary in L1, probably because such play activities are carried out mostly in L1.

Terminology

The term “expressive vocabulary” is used throughout this article, but not defined. It may be assumed that it is used to distinguish it from “receptive vocabulary”. However, in linguistics, the term “expressive" usually used to refer to elements expressing the speaker’s attitude, be it emotive or evaluative. The more common terms for “expressive” and “receptive” vocabulary as used in this article would be “active” and “passive” vocabulary.

Minor changes

The English language used in this article is generally acceptable, but there are certain aspects that require revision, for instance the word order within noun phrases. It would be a good idea to have a native speaker go over the entire text. The list below is not a complete list of what needs to be improved in terms of language.

Abstract:

l. 21 Remove double space.

l. 22: “We will discuss”: Change tense (to present), and consider using a passive construction. 

Main text:

- The use of the past tense in parts of this article, for example right at the beginning, is somewhat odd. The present tense is normally used in academic writing when referring to the content of an article.

l.31 “in most of the cases” > “in most cases”

l.45 “immigrant children 2-6 years old”: Ungrammatical; rephrase or change word order.

l.71  “no native” (l. 71) > “non-native”

l.78 “when parents address in L2 and L1”: The verb “address” is transitive; it requires an object.

l.80 “intended”: wrong word. Perhaps “defined as”?

l.84 “extra-family context” is a term created by non-native speakers that is misleading because it sounds like it is referring to an additional family; “non-family context” or “contexts outside the family” would be better choices.

l.95 “documented in each language a positive strong association”: Revise word order to “documented a strong positive association between ... in both L1 and L2.”

l.126 Insert a comma between “factor” and “given”

l. 134 “this last variable” > “the latter”

l.151 “child chronological age” > “a child’s chronological age”

l.241 “family’s education level” > “families’ education level”

l.332-333 “and only one child did so in L2”: This is unclear and seems to imply that only one child produced words in L2. “Did so” cannot refer to negated actions. Rephrase, perhaps as “whereas only one child did not produce any words in L2”.

l.336 “Forty-four”: If only percentages are given for the other groups, this absolute number should be omitted.

l.353 “for both L1 and L2”: I don’t think you mean for both languages, but rather “neither for L1 nor for L2”. (If you say “for both L1 and L2” you meant that there is an effect on both languages, not just one or the other.)

l.383, 387, 401, 412 What exactly is meant by “otherwise” here? Do you mean “however” or “on the other hand”?

l.443 “child’s” > “children’s”

l.453  “a positive, strong correlation” > “ a strong positive correlation”

l.454 “a positive moderate correlation” > “a moderate positive correlation”: Change word order throughout the article.

l.460-461: “When... exclusively.” This sentence has no verb; it should probably be linked to the following sentence with a comma.

l.466-567 and throughout the article “in the L1 child’s expressive vocabulary” > “in the child’s expressive L1 vocabulary”

l.485-486 “the type of parental language used in these settings may be qualitative or quantitative in terms of lexical and grammatical richness”: This makes little sense. How can the type of language be qualitative or quantitative? Rephrase.

l.491 “although the most frequently used compared to” > “although it is frequently used in comparison with”

l.531-532 “in literature” > “in the literature”

Summary

This article aims to supply new data, leading to new and original insights regarding an important topic (language acquisition by migrant children exposed). Generally, the methodology is adequate and the statistical analysis of the data is correct and appropriate for the objectives set out in this article.

The most serious flaw in this research is the source of the data. Rather than checking migrant children’s active and passive vocabulary in L1 and L2, parents were asked to tell, from memory, which words their children use or understand. This is an unreliable methodology with a potential to skew the results. Though the authors acknowledge this problem and attribute it to the difficulties triggered by the coronavirus pandemic, this lack of reliable data can and should be remedied now that the effects of the pandemic are no longer a hindrance. The otherwise good methodology used in this article is devalued by the unreliability of the data.

Another issue that could be improved is the article’s current lack of attention to the differences between the migrant families’ L1, and more importantly, the degree of similarity between their L1 and Italian (L2). The synergies between learning two closely related languages (e.g. Italian and Spanish or Italian and Romanian) are highly likely to affect bilingual language acquisition.

Despite this being a generally well-designed study that answers important question and advances our knowledge of the area, the flawed data collection process makes the results unreliable. For this reason, my recommendation is "reconsider after major revision", which refers, first and foremost, to a revision of the data collection process.

Reviewer 2 Report

I have read this paper with pleasure because it meets my scientific interests and I think it provides interesting and novelty results on individual and contextual (home language environment; extra-familial environment) factors and expressive vocabulary in immigrant preschoolers. I would like to give the authors some suggestions to further improve this work.

Introduction

The authors considered several factors, mainly contextual but also individual factors. These factors are not introduced at the beginning of the Introduction but are presented throughout the Introduction. I suggest the authors to better structure the Introduction to clarify the considered factors. Specifically, 

-       The authors may want to refer to the bioecological model of development (Bronfenbrenner & Morris, 1998), used also by Erika Hoff (2006), to delineate the contextual factors that affect language acquisition. 

-       Lines 61-118. The authors provided clear definitions of measures of quantity and quality of language exposure but the distinction between measures of home exposure and exposure in the extra-familial context is not clear at the beginning of this paragraph. I suggest the authors include this relevant distinction from the beginning. Based on my reading, I think the authors aimed at disentangling the relative contribution of exposure at home and the contribution of extra-familial contexts; if this is the case, I suggest the authors slightly modify the last sentence (lines 117-118) to clarify this point.

Bronfenbrenner, U. & Morris, P. A. (1998). The ecology of developmental processes. In W. Damon (Series Ed.) & R. M. Lerner (Vol. Ed.), Handbook of child psychology, Vol. 1: Theoretical models of human development (5th ed., pp. 993–1028). Wiley.

Hoff, E., (2006). How social contexts support and shape language development. Developmental Review, 26, 55–88. https://doi.org/10.1016/j.dr.2005.11.002

Lines 119-166. At first reading, it is not clear what this paragraph adds compared to the previous one which is focused on the associations between the variables of interest. Based on my reading, I think the studies reviewed in this paragraph analyzed the “unique contribution” of various measures of language exposure, familiar factors, and individual factors; this point should be clarified. Relatedly, the authors distinguished between associations and predictors; I guess this distinction is based on the statistical analyses used in the studies (e.g., correlations vs regressions). I suggest the authors justify the terminology used or, as stated above, speak about "unique contributions or contributors".

Line 179: The distinction between HLE-active and passive activities is clear but I suggest the authors clarify why HLE-play was considered separately compared to HLE active and passive (this is mentioned in the Method, lines 288-291, but should be better justified in the Introduction).

Materials and Methods

Materials and Methods are described clearly. I suggest the authors to:

-   clarify how the percentages of parents’ Italian oral fluency were calculated (lines 295-296)

-   describe in more detail the analyses in the “Plan of the analyses” (lines 310-317); in the current version, the analytic plan is described partly in this paragraph and partly in the Results section (e.g., lines 367-378; lines 391-399..).

Results

Table 1: what does the asterisk reported for L2 parental oral fluency refer to?

Table 2: what does “ra” refer to?

Discussion

The authors considered and discussed point-by-point the new contributions of the study. I suggest the authors further elaborate on the Discussion. Specifically, I suggest to:

-       focus on the new contribution of their work rather than reviewing previous literature (e.g., lines 470-478)

-       slightly modify parts in which they comment on factors that did not significantly contribute to expressive vocabulary (e.g., lines 479-481; 498-499); I can see that analyzing the unique contributions of several factors (e.g., HLE-passive and play) is the new contribution of this work but is rather unusual to comment on that the “lack of predictive role” is an original contribution.

-       Line 485-486; this sentence should be clarified.

-       Lines 494-495 and 541-542; the authors commented on HLE-passive findings on L1 and L2 expressive vocabulary (e.g. “In addition, we hypothesize that given the age and level of L1 of our immigrant preschoolers, the type of language used on TV is not fully understood and is not able to  influence their L1 expressive abilities”). This explanation should be clarified because it took into account the characteristics of TV language input (Why is TV language difficult?) and children’s comprehension skills (Do the authors can provide some descriptive data, based on observation of the participants or parents’ report to support this statement?)

Round 2

Reviewer 1 Report

I reluctantly accept the use of MB-CDI because parent reporting is used in numerous studies of this kind, though my own experience tells me that parents' knowledge of their children's linguistic performance is unreliable. Also, there are publications casting doubt on the validity of MB-CDI, e.g.:

Eriksson, Mårten. 2022. "Insufficient evidence for the validity of the Language Development Survey and the MacArthur–Bates Communicative Development Inventories as screening tools: A critical review". International Journal of Language & Communication Disorders 2022: 1-21.

I also still believe that some basic quantification of the degree of similarity between L1 and L2 should have been included as a potential factor. Though the authors rightly point out that the article focuses on the effects of quantity and quality of exposure, it is likely that these effects are different for children whose L1 and L2 are lexically similar.  I would very much advise to include this parameter in future research.

Please revise the tenses. In the last sentence of the abstract, "has been discussed" makes little sense; the present tense ("is discussed") should be used here.

Author Response

I thank Reviewer 1 for his/her suggestions.
I will take his/her comments into consideration in future research.
I changed the verb tense in the abstract section.

Arianna Bello